# Chayote (*Sechium edule* (Jacq.) Swartz) Seed as an Unexploited Protein Source: Bio-Functional and Nutritional Quality of Protein Isolates

**DOI:** 10.3390/foods12152949

**Published:** 2023-08-04

**Authors:** Elsa F. Vieira, Ana Q. Fontoura, Cristina Delerue-Matos

**Affiliations:** REQUIMTE/LAQV, ISEP, Polytechnic of Porto—School of Engineering, Rua Dr. António Bernardino de Almeida, 431, 4249-015 Porto, Portugal; 1181232@isep.ipp.pt (A.Q.F.); cmm@isep.ipp.pt (C.D.-M.)

**Keywords:** chayote seed, protein isolate, ultrasound-assisted extraction, functional properties, nutraceutical, in vitro digestibility

## Abstract

Chayote seeds have good protein quality and recognized bioactive properties, being still unexplored as a nutraceutical. In this work, chayote seed protein isolates (CSPIs) were prepared by alkaline extraction (AE) and ultrasonic-assisted extraction (UAE) using a probe (20 kHz) or a water bath (40 kHz), and their physicochemical, functional properties and nutraceutical potential were investigated. For all treatments, protein solutions (10% *w*/*v*) were treated for 20 min. The UAE significantly (*p* < 0.05) improved the protein extraction yield and functional properties (protein solubility, turbidity, and emulsifying and foaming properties) of CSPIs. This effect was more pronounced using a probe sonication device. The CSPI obtained by UAE-20 kHz contained 8.2 ± 0.9% dw of proteins with a balanced amino acid profile, higher content of essential amino acids (315.63 mg/g of protein) and higher protein digestibility (80.3 ± 4.5%). Furthermore, CSPI.UAE-20 kHz exhibited the highest phenolic content (7.22 mg GAE/g dw), antioxidant capacity and α-amylase inhibition (74%, at 100 μg/mL concentration). Overall, these results suggest that ultrasound technology contributed greatly to the corresponding functional and nutritional properties of chayote seed proteins. It would be, therefore, useful to apply this Cucurbitaceae species in food systems, promoting its nutritional and commercial value.

## 1. Introduction

Plant proteins from food matrices and by-products are getting great interest due to the high demand for alternative sources to animal proteins, which are generally associated with a negative environmental impact and high production costs [1,2]. Several protein isolates and hydrolysates from different vegetable sources have been extensively studied; the alkaline or acid extraction/precipitation methods have been the conventional extraction methodologies employed [3]. Nevertheless, the use of extreme extraction conditions could reduce the nutritive value of protein, degradation of bioactive compounds, and promote undesirable functional properties. To overcome this drawback, novel and sustainable approaches have been investigated, namely enzymatic, ultrasound, microwave, pulsed electric field, and supercritical fluid extraction [4]. In this field, ultrasound-assisted extraction (UAE) has arisen as a promising environmentally friendly technology to increase the extraction efficiency of proteins from vegetable sources, with preservation of bioactive material and improvement of physicochemical properties of proteins, such as solubility, turbidity, emulsification, foaming, and thermal properties [5,6]. UAE technique is based on the propagation of pressure oscillations in a liquid medium at the speed of sound, which results in the formation, growth, and collapse of microbubbles, allowing cell disruption and a mass transfer to the medium [7]. UAE is energy efficient, easy to install, with minimal environmental impact, and its maintenance costs are low and requires low investment and shorter extraction times, thus reducing the processing time and the associated costs [8].

Several protein isolates and hydrolysates of vegetable origin have been obtained by UAE, with potential use as functional ingredients in diverse food formulations, such as bakeries, meats, sauces, candies, cosmetics, snacks, and non-carbonated beverages [9]. This extraction technique has been efficiently applied to produce protein isolates and hydrolysates from certain Cucurbitaceae species. Naik et al. (2022) [3] compared alkali extraction and pulsed ultrasonic-assisted extraction (PUSAE) to extract protein from bitter melon seeds, showing that the highest yield resulted from PUSAE, which also improved some techno-functional properties (water holding capacity, solubility, emulsion capacity, emulsion stability, and gelation capacity) of the extracted proteins. Similarly, Du et al. (2022) [9] applied UAE to evaluate changes in the physicochemical, structure, and foaming properties of pumpkin seed protein isolates. It was shown that UAE contributed greatly to the functional and nutritional values of this vegetable protein, improving its added value. Likewise, Wen et al. (2019) [8] evaluated the use of UAE as a pre-treatment of watermelon seed protein isolates to produce antioxidant hydrolysates, showing that ultrasounds strongly influenced the structure and enzymatic efficiency of watermelon seed protein isolate, contributing to enhanced antioxidant activity and stability of protein hydrolysates. 

Chayote (*Sechium edule* (Jacq.) Swartz) belongs to the Cucurbitaceae family, among pumpkin cucumber and bitter melon, and has gained widespread consuming acceptance and is recognized by its nutritional and bio-functional properties. This Cucurbitacea species grows in tropical, subtropical, and warm regions around the world and each plant can produce 80–100 fruits, reaching production yields of 54,000 lb/ac [10]. Fruits have been traditionally used in the food industry to produce purees, juices, jams, and alcoholic beverages [11]. Compared to other Cucurbitaceae seeds, such as bitter melon (28–31% dw) [3], pumpkin seed (35.18% dw) [2], or watermelon (30.63–43.60% dw) [8], the protein content of chayote seeds is relatively low (around 5.50% dw) [11]. However, the protein fraction is rich in essential amino acids (EAA), which indicates good protein quality. Moreover, chayote seeds present a low lipid content (less than 1% dw) [11], being a positive aspect to improve protein extraction efficiency. Besides the nutritional potential, chayote seeds have been claimed to possess antioxidant and antimicrobial properties [11]. Little is known regarding the biochemical characterization, functional, and nutritional properties of seed proteins of chayote, and to our best knowledge, no previous studies have been conducted on protein extraction from chayote seeds and the preparation of protein isolates. Considering that chayote seeds proteins have good quality, the obtained isolates and extracts could be rich in bioactive peptides with promising bio-functional properties to be explored in food systems. Taking this into consideration, this study was carried out to investigate the effects of alkaline and ultrasound (comparing water bath and probe devices) treatments on the recovery of protein from chayote seeds. The protein isolates were characterized regarding some bioactive and functional properties, as well as protein bioaccessibility. The possible outcome of this research is the innovative utilization of chayote seeds as a source of proteins, which would increase the application of this Cucurbitaceae in food systems.

## 2. Materials and Methods

### 2.1. Reagents

Folin–Ciocalteu, 2,2-azino-bis-3-ethylbenzothiazolin-6-sulfonic acid (ABTS), bovine serum albumin (BSA), pancreatic α-amylase (Sigma-A3176), porcine gastric mucosa pepsin (Sigma-P7012), porcine pancreas pancreatin (Sigma-P7545), porcine bile extract (Sigma-B8631), soybean oil, ferric chloride, trichloroacetic acid (TCA), and a standard mixture of eight MW markers (6.5–66 kDa) were purchased from Sigma Aldrich (Steinheim, Germany). Gallic acid was obtained from Fluka (Steinheim, Germany) and sodium hydroxide was from Labkem (Barcelona, Spain). Ultrapure water resistivity of 18.2 MΩ cm was from a Simplicity 185 water purification system (Millipore, Molsheim, France). All the spectrophotometric assays were performed in a Synergy HT W/TRF Multi Mode Microplate Reader with Gen5 2.0 software (BioTek Instruments, Winooski, VT, USA).

### 2.2. Material 

Chayote fruits were supplied by a local farm located at Cinfães (41.08395609505081, −8.070946173549956), Douro, Portugal. The mature fruits were collected in October 2022 from 15 plants to obtain a representative set of fruits. Fruits were examined for integrity and cleaned with tap water. Seeds were separated from the flesh and dried for 8 h at 52 °C in processed food (Excalibur 9 Tray Dehydrator, Model 4926 T, Sacramento, CA, USA) to <11% moisture content. The dried seeds were grounded (Moulinex A320, Paris, France) to fine powder, sieved with 0.75 mm stainless steel and defatted twice with hexane (1:5 *w*/*v*) for 5 min at room temperature. The defatted samples were then dried and stored at −20 °C until further analysis. The contents of moisture and protein of chayote seed powder (CSp) were 14.83 ± 1.38% and 5.5 ± 0.4%, respectively, based on the official AOAC methods of analysis [12]. 

### 2.3. Preparation of Chayote-Seed Protein Isolates (CSPI)

Three approaches were assessed to produce protein isolates (CPSIs) from chayote seed powder (CSp): alkaline extraction, ultrasonic-assisted extraction using a probe device, and ultrasonic-assisted extraction using a water bath (Figure 1). 

For alkaline extraction, CSp was suspended in distilled water (10%, *w*/*v*), the pH was adjusted to 11 by the addition of 2 M NaOH and the mixture was stirred for 20 min at temperature below 10 °C. After pH adjustment to 4.0 (by addition of 1 M HCl) and centrifugation at 4 °C, 10,000× *g* for 10 min, the recovered protein precipitate was washed twice with distilled water. For ultrasonic-assisted extraction, CSp was suspended in distilled water (10%, *w*/*v*), and the pH was adjusted to 11 by the addition of 2 M NaOH and sonicated for 20 min using an ultrasound probe or an ultrasound bath. For probe sonication, 50 mL of CSp dispersion was sonicated in a 100 mL conical flask, which was immersed in an ice bath to maintain the temperature below 10 °C, at a frequency of 20 kHz (500 W and 25% amplitude). Extraction was performed by an ultrasonic processor (Sonic Vibracell, model VC 750, Newtown, CT, USA), comprising a 13 mm diameter tip. For bath sonication (Selecta SA, Barcelona, Spain), 50 mL of CSp dispersion was placed in a 100 mL flat bottom conical flask and sonicated at frequency of 40 kHz and 500 W power. Ice was added to bath to maintain the temperature below 10 °C. Like alkaline extraction, the pH of sonicated samples (from probe and water bath devices) was adjusted to 4.0 with 1 M HCl, and the protein fraction was recovered from the liquid phase by centrifugation at 4 °C and 10,000× *g* for 10 min and washed twice with distilled water. The protein isolates obtained by alkali extraction, probe, and water bath sonication, respectively coded as CSPI.AE, CSPI.UAE-40 kHz and CSPI.UAE-20 kHz were freeze-dried (Telstar, model Cryodos −80, Terrasa, Spain) and stored at −20 °C for subsequent analyses. 

### 2.4. Protein Extraction Yield (%) and Protein Purity (%)

The protein content of chayote seed powder (CSp) and protein isolates (CSPIs) were determined in duplicate by adopting the modified Kjeldahl procedure [12]. Samples digestion was performed in a DK6 Heating digester (Velp Scientific, Usmate, Italy) and distilled in a Keltec System 1002 Distilling Unit (Foss Tecator, Hilleroed, Denmark). The conversion factor was taken as 5.30 [7]. Protein extraction yield (%) and protein purity (%) were calculated by Equations (1) and (2):(1)$$\mathrm{Protein}\;\mathrm{extraction}\;\mathrm{yield}\left(\%\right)\frac{amount\;of\;protein\;in\;CSPI\;(\frac{mg}{g})}{amount\;of\;protein\;in\;CSp\;(\frac{mg}{g})}\times \;100,$$
(2)$$\mathrm{Protein}\;\mathrm{purity}\;\left(\%\right)\frac{amount\;of\;protein\;in\;CSPI\;(\frac{mg}{g})}{CSPI\;weight\;(g)}\times \;100.$$

### 2.5. Size Exclusion Chromatography (SE-HPLC)

The molecular weight (MW) distribution of CSp and CSPIs was analysed by Size Exclusion Chromatography (SE-HPLC) according to Vieira et al. (2017) [13] methodology. The column used was a PSS Proteema Analytical 100 Å column (Amersham Biosciences, Buckinghamshire, UK), equilibrated with 50 mM sodium phosphate buffer, 0.15 M NaCl, pH 6.6 at a flow rate of 0.3 mL/min and calibrated using a standard mixture of eight MW markers: albumin (66 kDa), ovalbumin (45 kDa), glyceraldehyde-3-phosphate dehydrogenase (36 kDa), carbonic anhydrase (29 kDa), trypsinogen (24 kDa), trypsin inhibitor (20 kDa), α-lactalbumin (14.2 kDa) and aprotinin (6.5 kDa). For the chromatographic analysis, lyophilized CSPIs were dissolved in the mobile phase (at the protein concentration of 1.0 mg/mL) and an injection volume of 20 µL was used. Peak signals were detected at 280 nm with SPD-M20A photodiode array detector. Analyses were performed in triplicate. Results were expressed as percentages on groups of different MW: (i) >66 kDa, (ii) 66 to 45 kDa, (iii) 45 to 24 kDa, (iv) 24–6.5 kDa, (v) <6.5 kDa.

### 2.6. Functional Properties

#### 2.6.1. Protein Solubility

The solubility of CSp and CSPIs was assessed at pH ranging from 2 to 12, following Achouri et al. (2012) [14] methodology, with some modifications. Samples (2 mg/mL) were dispersed in distilled water, and pH was adjusted to 2, 4, 6, 8, 10, and 12 by the addition of (1 M or 6 M) HCl or NaOH. The mixture was stirred occasionally for 60 min at 4 °C and centrifuged at 7500× *g* for 15 min and the protein contents of supernatants were determined by the Bradford assay (2011) [15], using BSA as standard. Protein solubility was calculated by Equation (3): (3)$$S\left(\%\right)=\frac{Amount\;of\;protein\;in\;supernatant}{Amount\;of\;protein\;in\;dispersion}\times \;100.$$

#### 2.6.2. Turbidity

The turbidity analysis of CSp and CSPIs was based on Malik et al. (2017) [5] methodology. Samples (2 mg/mL, *w*/*v*) were stirred for 60 min at room temperature, and then the absorbance was measured at 600 nm.

#### 2.6.3. Emulsification Properties

Emulsifying capacity (EC) and emulsion stability (ES) of CSp and CSPIs were determined following Lawal et al. (2007) [16] procedure. Samples (10 mg/mL) were dispersed in distilled water and adjusted to pH 4, 6, 8, 10, and 12. Then, 5 mL of sample solution were homogenized with 5 mL soybean oil and after centrifugation at 3500 rpm for 5 min, the height of emulsified layer and the height of the total contents in the tube was measured. ES was determined by heating the emulsion at 80 °C for 30 min before centrifuging at 3500 rpm for 5 min. EA and ES were calculated by Equation (4): (4)$$EC\left(\%\right)or\;ES\;(\%)=\frac{Height\;of\;emulsified\;layer}{Height\;of\;total\;content\;in\;the\;tube}\times \;100.$$

#### 2.6.4. Foaming Properties

The foaming properties were determined according to Du et al. (2022) [9] procedure, with slight modifications. Briefly, 10 mL of sample solution (10 mg/mL) at pH 2, 4, 6, 8, 10, and 12 were whipped at high speed in a mixer blender for 5 min to produce foam. After that, the foam was immediately poured into a 25 mL graduated cylinder. Foaming capacity (FC) was determined by comparing the foam volume at 2 min with the initial liquid volume of the sample (Equation (5)), while the foam stability (FS) was measured by comparing the foam volume at 20 min with the initial foam volume (Equation (6)): (5)$$FC\left(\%\right)=\frac{V_{0}}{25}\times \;100,$$
(6)$$FS\left(\%\right)=\frac{V_{1}}{V_{0}}\times \;100,$$
where V_0_ represents the volume of foam at 2 min, and V_1_ represents the volume of foam at 20 min. 

### 2.7. Nutraceutical Potential

#### 2.7.1. Amino Acid Composition

Hydrolysis of total protein and the analysis of amino acid composition was performed in duplicate according to Vieira et al. (2018) [17] methodology. The liquid chromatograph consisted of a Shimadzu LC system (Shimadzu Corporation, Kyoto, Japan) equipped with an LC-20AD pump, DGU-20AS degasser and photodiode array SPD-M20A (PAD), and fluorescence RF-10AXL (FLD) detectors. The relative amino acid composition was expressed as mg/g protein. 

#### 2.7.2. Protein Digestibility

The in vitro digestion of CSp and CSPIs was assessed by the internationally standardized protocol of the simulated gastrointestinal digestion process described by Minekus et al. (2014) [18] with slight modifications. Oral (1 g of each sample was mixed with 4 mL of simulated salivary fluid (SSF) containing 0.5 mL of α-amylase (75 U/mL) at pH 7 for 2 min of incubation), gastric (each oral digest was mixed with 8 mL of simulated gastric fluid (SGF) containing pepsin (2000 U/mL) at pH 3 for 2 h of incubation), and intestinal (each gastric digest was mixed with 8 mL of simulated intestinal fluid (SIF) containing pancreatin (100 U/mL) and bile (10 mM) at pH 7 for 2 h of incubation) digestive mixtures were incubated in a water bath at 37 °C and constant shaking (100 rpm). SSF, SGF, and SIF solutions were prepared according to Minekus et al. (2014) [18]; blanks were performed just with digestive enzymes and all samples were blank corrected. The digests were cooled by immersion in an ice bath and then centrifuged at 5000× *g* for 10 min at 4 °C to separate the soluble bioaccessible fraction from the residual fraction. Supernatants from the bioaccessible fraction were subsequently frozen at—20 °C. until analysis. Protein digestibility was calculated by Equation (7):(7)$$\mathrm{In}\;\mathrm{vitro}\;\mathrm{Protein}\;\mathrm{Digestibility}\;\left(\%\right)=\frac{\mathrm{bioacessible}\;\mathrm{protein}\;(\mathrm{mg}/g)}{\mathrm{initial}\;\mathrm{proten}\;\mathrm{content}\;(\mathrm{mg}/g)}\times \;100$$

#### 2.7.3. Nutritional Protein Quality 

The essential amino acid profile (EAA) was used to determine the protein nutritional quality of CSp and CSPIs. The EAA scores (EAAS) and EAA index (EAAI) were calculated based on Friedman (1996) [19], using Equations (8) and (9): (8)$$\mathrm{EAAS}=\frac{a_{p}}{a_{s}},$$
(9)$$EAAI\left(\%\right)=\sqrt[{n}]{\frac{{Lys}_{1p}}{{Lys}_{1s}}\times \frac{{Tyr}_{2p}}{{Tyr}_{2s}}\times \ldots \times \frac{{His}_{np}}{{His}_{ns}}}\times \;100,$$
where a is an EAA, *p* is the test protein, *s* is the FAO/WHO/UNU (2007) [20] reference protein, and *n* is the number of amino acids included into the calculation. 

The protein digestibility corrected amino acid score (PDCAAS) was calculated based on FAO/WHO/UNU (2007) [20], using Equation (10): (10)PDCAAS=Lowest uncorrected amino acid score× IVPD
where IVPD is the in vitro protein digestibility (%).

#### 2.7.4. Total Phenolic Content

Determination of total phenolic content (TPC) was based on the Folin-Ciocalteau assay as described by Paz et al. (2015) [21] using gallic acid as standard. Briefly, 10 mg of lyophilized CSp and CSPIs were diluted in 10 mL of distilled water. Then, 25 μL sample was mixed with 25 μL Folin-Ciocalteau reagent, 75 μL of distilled water and 100 μL of 75 g/L Na_2_CO_3_. After keeping at 40 °C for 90 min, the absorbance was measured at a wavelength of 765 nm. The results were expressed as mg gallic acid equivalents (GAE) per gram of sample. All measurements were carried out in triplicates.

#### 2.7.5. Antioxidant Capacity 

The reducing power (RP) of CSp and CSPIs was measured according to Vieira et al. (2016) [22] procedure with slight modifications. Briefly, 250 μL of sample suspension (25, 50, and 100 μg/mL, prepared in distilled water) were mixed with 250 μL of sodium phosphate buffer (200 mM, pH 6.6) and 250 μL of potassium ferricyanide 1% (*w*/*v*). The mixture was incubated at 50 °C for 20 min. Next, 250 μL of cold TCA 10% (*w*/*v*) was added to stop the reaction. Afterwards, the mixture was centrifuged at 650 rpm for 10 min. Then, 500 μL of supernatant was incubated with 500 μL of deionized water and 100 μL of ferric chloride 0.1% (*w*/*v*). After 10 min reaction, the absorbance was measured at 700 nm. RP values were expressed as μg/mL and calculated by Equation (11):(11)$$Reducing\;power=\frac{A_{1}}{A_{0}}\times \;100,$$
where A_0_ is the absorbance of blank and A_1_ is the absorbance of sample suspensions after 10 min of incubation.

The 2,2′-azino-bis (3-ethylbenzothiazoline-6-sulphonic acid) radical scavenging activity (ABTS) of CSp and CSPIs was determined according to Gião et al. (2007) [23]. Briefly, 20 μL of sample suspension (25, 50, and 100 μg/mL, prepared in distilled water) or control (distilled water) were mixed with 980 μL ABTS^•+^ solution and incubated in the darkness at 30 °C for 10 min (completion of the reaction time), followed by absorbance reading at 734 nm in a microplate reader. Each experiment was performed in triplicates and scavenging activity was calculated by Equation (12):(12)$$ABTS\;radical\;scavenging\;activity\left(\%\right)=\frac{\left(A_{0}\;-\;A\right)}{A_{0}}\times \;100,$$
where A_0_ is the absorbance of blank ABTS radical solution and A is the absorbance of sample suspensions after 10 min of incubation. 

#### 2.7.6. Anti-Diabetic Activity

The porcine pancreatic α-amylase inhibition assay was based on Esfandi et al. (2022) [24] protocol, with slight modifications. Briefly, different concentrations of CSPIs suspensions (25, 50, and 100 µg/mL) were prepared in PBS solution (0.02 M sodium phosphate buffer, pH 6.9, containing 6 mM sodium chloride). Afterwards, 10 μL of α-amylase (15 U/mL in PBS solution) were added to 500 μL of each CSPI suspension and mixtures were incubated for 10 min at room temperature, followed by the addition of 500 μL of soluble starch (1%, *w*/*v* prepared in PBS solution) and incubation for 20 min. Then, 100 μL of 1 N dilute hydrochloric acid was added to stop the enzymatic reaction, followed by the addition of 200 μL of freshly prepared iodine solution. Absorbance was read at 680 nm.

### 2.8. Statistical Analysis

Data were subjected to analysis of variance (ANOVA) using SPSS software (SPSS 20.0, Chicago, IL, USA), followed by Duncan’s multiple range test with 95% confidence (*p* ≤ 0.05). 

## 3. Results and Discussion

### 3.1. Protein Extraction Yields and Protein Purity

According to Table 1, the protein content of CSp was 5.5 ± 0.4%; and 6.6. ± 0.6%, 8.2 ± 0.9% and 7.7 ± 1.3% for CSPIs obtained by alkaline and ultrasonic treatments, respectively. CPSI.UAE-20 kHz presented a significant (*p* < 0.05) higher protein content and protein yield recovery (88.0 ± 3.2%), which suggests that the higher turbulence created by ultrasonic cavitation probe at a frequency of 20 kHz (500 W and 25% amplitude) was more effective in disrupting the chayote seed material and favoring the penetration of extraction solvent to improve mass transfer. In addition, the ultrasonic probe device (20 kHz) favored the direct contact between tip and sample, which do not occur in the water bath apparatus (40 kHz), where samples do not contact directly with irradiating surface. Thus, CSp treated with 20 kHz probe is more exposed to high power compared to 40 kHz water bath, leading to greater extraction efficiency. Moreover, the ultrasound treatment produced CSPIs with higher (*p* < 0.05) protein purity, the values found were 66.7 ± 2.5% and 64.4 ± 5.1 for CSPI.UAE-20 kHz and CSPI.UAE-40 kHz, respectively.

### 3.2. Molecular Size Distribution of CSPIs 

The molecular weight (MW) profiles and distribution of CSp and CSPIs, obtained by alkaline and ultrasound treatments, are shown in Figure 2. CSp presented large molecular mass proteins (~50%) with MW higher than 66 kDa and a less fraction (~20%) with 29–66 kDa. This last fraction showed two prominent fractions with 27 kDa and 48 kDa. The 27 kDa protein could be sechiumin, a ribosome-inactivating protein (RIP), with claimed chemotherapeutic activity [25], and the 48 kDa protein could be β-glucosidase [26]. Moreover, the CSp molecular weight pattern also agrees with Vladova et al. (2004) [27], which described cucurbitin, a globular protein with MW of 54 kDa, as the main storage protein found in Cucurbitaceae seeds. Compared with untreated chayote seed proteins (CSp), the formulated protein isolates (CSPIs) showed significant (*p* < 0.05) changes in the chromatographic profiles, indicating that alkaline and UAE treatments were effective in reducing the molecular size of chayote seed proteins. As depicted in Figure 2B, different MW distribution profiles were obtained for CSPI.AE, SPI.UAE-20 kHz, and SPI.UAE-40 kHz. UAE with a prove device (20 kHz, 500 W, 25% amplitude) produced protein isolates with significantly (*p* < 0.05) higher content of peptides with MW below 6.5 kDa (~31%). This fraction was ~25% for CSPI produced using an ultrasound bath (40 kHz, 500 W) and ~8% when chayote seed proteins were alkali extracted. CSp treated with a 20 kHz probe is more exposed to high power compared to a 40 kHz water bath, leading to greater hydrolysis of proteins with the formation of smaller peptides.

### 3.3. Functional Properties of CSPIs

#### 3.3.1. Protein Solubility 

Solubility is a good index of protein functionality and is influenced by pH changes [2]. This property is a practical measurement of protein denaturation and aggregation, being related to other functional properties, namely emulsification, foaming and gelation, and has an influence on color, texture, and sensory quality of products [1,3,5]. According to Figure 3, CSPIs obtained from the three extraction treatments presented higher (*p* < 0.05) solubility than CSp at varied pH values.

The solubility of CSPIs presented a typical U-shaped curve within the controlled pH range of 2–12. Lower solubilities were estimated at pH value 4, the isoelectric point region for proteins, while maximum solubilities were observed at a pH value of 10. At pH 10, SPI.UAE-20 kHz presented a significantly higher (81.0% ± 1.8, *p* < 0.05) solubility than SPI.UAE-40 kHz (74.1% ± 3.0) and CSPI.AE (67.3% ± 2.1), suggesting that UAE by a prove (20 kHz, 500 W, 25% amplitude) is more favorable in reducing the particle size of CSp and enhancing the protein–water interactions, which contributed to the formation of more soluble protein aggregates. Similar observations were found for other vegetable seed proteins submitted to ultrasound treatments, such as rapeseed [28] and sunflower meal [5], as well as for other Cucurbitaceae species, like bitter gourd [3], watermelon [8], and pumpkin [2]. 

#### 3.3.2. Turbidity 

The turbidity is related to particle size, proteins, and structure [9] and was quantified as the percentage of transmitted light measures at 600 nm. The values found were 0.92 ± 0.02, 0.71 ± 0.01, and 0.75 ± 0.01 for CSPI.AE, CSPI.UAE-20 kHz, and CSPI.UAE-40 kHz, respectively. Less turbid protein solutions were obtained after ultrasound treatment in comparison to alkaline extraction, which indicates the formation of small particles since there is less light scattering. This observation is in line with the lowest fraction found in the Size exclusion chromatographic analysis (Figure 2). Similar observations were reported by Malik et al. (2017) [5] for defatted and dephenolized sunflower meal and by Du et al. (2022) [9] for pumpkin seed protein isolates. These authors found that a decrease in turbidity after ultrasonication was attributed to a disruption in protein–protein interactions, which prevented the formation of large aggregates and contributed to the formation of small aggregates, resulting in a less turbid solution.

#### 3.3.3. Emulsifying Property

Emulsifying capacity (EC) is the ability of a protein to form an emulsion, while emulsion stability (ES) refers to the capability of emulsion droplets to remain dispersed without coalescing, flocculation, or creaming [2]. Both properties are important for several food applications (e.g., chopped meat, cake batters, salad dressings, mayonnaise, and frozen desserts) and are strongly affected by the MW, hydrophobicity, solubility, conformation stability, charge, pH, ionic strength, and temperature [29]. Figure 4 shows the emulsifying capacity and emulsion stability observed for the CSp and CSPIs at a pH range of 2–12. 

Both properties exhibited U-shape curves. As observed for protein solubility (Figure 3), for all CSPIs the lowest value of emulsion capacity was found at pH 4, close to the isoelectric point of chayote seed proteins, while the highest values were obtained at pH 12. A comparable trend was reported for other proteins isolated from sunflower seed meal [7], camelina, and flixweed seed meals [1]. At pH 10, the CSPI.UAE-20 kHz showed the highest (*p* < 0.05) emulsion capacity (51 ± 3.1%), followed by CSPI.UAE-40 kHz (44% ± 7.1%) and CSPI.AE (31% ± 3.2%), Figure 4A. Regarding the emulsion stability results, it was observed that chayote seed protein isolates produced by ultrasound treatment presented significantly higher emulsion stability (around 58%, *p* < 0.05) than proteins isolated by the alkaline treatment (45 ± 2.1%), Figure 4B. These results suggest that UAE approaches (probe or water bath) seem to affect the structure of chayote seed proteins by enhancing their molecular flexibility and surface hydrophobicity. A positive effect of UAE on emulsion capacity and stability was also reported for proteins isolated from sunflower seed meal [5], peanut [30], camelina and flixweed seed meal [1], and bitter melon seed [3]. Moreover, the emulsifying capacity of CSPIs were in the same range as those found for sunflower meal protein isolates [7]. The emulsion stability of CSPIs was however higher than that observed for pumpkin seed protein isolate extracted at pH 9 with 1.0 M NaCl [2]. 

#### 3.3.4. Foaming Properties

Foaming capacity (FC) and foam stability (FS) are strongly correlated with protein solubility since higher water–protein interactions help to unfold the protein structure, enhancing the air encapsulation [1]. As depicted in Figure 5, significant difference at *p* < 0.05 were obtained in the foaming properties of CSp and CSPIs at a pH range of 2–12. For all samples, the lowest foaming capacity and foaming stability were found at pH 4, which is the point of least protein solubility (Figure 3) and increased in the pH range of 4–12.

At several pH conditions, the foaming capacity of CSPIs obtained by ultrasounds treatment was significantly (*p* < 0.05) higher than those of untreated chayote seed proteins (CSp). For instance, at pH 10, the foaming capacity of CSPI.UAE-20 kHz (71% ± 2.1) and CSPI.UAE-40 kHz (63% ± 3.3) were significantly higher (*p* < 0.05) than values observed for the alkaline extracts (54% ± 2.0). As shown in Figure 5B, the foam stability of CSp was also improved after ultrasound treatment; at pH 10, values were 82% ± 1.8, 70% ± 3.0 and 70% ± 2.1 for CSPI.UAE-20 kHz, CSPI.UAE-40 kHz, and CSPI.AE, respectively. The results indicated that UAE performed with a probe device, promoted a surface hydrophobicity increase of chayote seed proteins, leading to a higher foam formation and stability. These results corroborate with those obtained by Morales et al. (2015) [31] for soybean protein isolates and by Du et al. (2022) [9] for pumpkin seed protein isolates when high-intensity ultrasonic conditions (20 kHz, 500 W, for 20 min) was applied. Authors stated the decreased particle size within the ultrasound treatment could lead to the increased adsorption of protein at air–water interface, thus improving the foaming properties of protein isolates. 

### 3.4. Nutritional Properties of CSPIs

#### 3.4.1. Amino Acid Composition 

Table 2 presents the amino acid composition of CSp and CSPIs, obtained by alkaline and ultrasound treatments. Data showed that CSp protein is rich in Arg (135.40 mg/g protein), Glu (65.70 mg/g protein), and except for Leu presents all the EAAs, indicating a good protein quality. Data are supported by Flick et al. (1978) [32], who described Arg, Glu as the highest amino acid concentration within the profile of chayote seeds, although Leu was also quantified in representative concentrations. Differences in the genetic and geographic diversity of chayote could explain the variations in the amino acid profile of chayote seeds verified in this work. 

From Table 2, it is also observed that the preparation of protein isolates from CSp had an impact on the amino acids profile, with significant differences (*p* < 0.05) for all amino acids, except His, Lys, Val, Met, and Pro. Compared to untreated chayote seed, CSPIs obtained by UAE treatment using a probe or water bath devices were particularly rich in Arg (with respective mean contents of 167.46 and 166.95 mg/g protein for CSPI.UAE-20 kHz and CSPI.UAE-40 kHz), Glu (87.37 and 83.37 mg/g protein), Leu (58.91 and 54.06 mg/g protein), and Asp (57.83 and 54.32 mg/g protein). Vinayashree and Vasu (2021) [2] also reported high contents of these amino acids for pumpkin seed protein isolates obtained at pH 9, 1.0 M NaCl, and the flour-to-solution ratio of 1:25 (*w*/*v*). The richness in Arg contents suggests that CSPIs could be beneficial to treat cardiovascular disease [33]. Data also showed significant differences (*p* < 0.05) in the contents of hydrophobic amino acids (Val, Ile, Leu, and Phe) of CSp and CSPIs. The presence of high levels of hydrophobic amino acid residues in CSPI.UAE-20 kHz and CSPI.UAE-40 kHz could be responsible for a more compact interior core of the protein and, thus, for a more stable protein. Nevertheless, the levels of hydrophobic amino acids found for CSPIs were in the lower range (31.30–32.32%) than that reported by Vinayashree and Vasu (2021) [2] for pumpkin seed protein isolates (39.28%). Compared to alkaline protein isolate, the ultrasound CSPIs presented significantly (*p* < 0.05) higher concentrations of almost all amino acids, without affecting the % of EAA, which was around 35% for all treatments. The quality of a dietary protein can be assessed by comparison with the FAO/WHO/UNU (2007) recommended pattern of essential amino acids (EAAs). CSp and CSPIs presented EAA values within or above the FAO/WHO/UNU (2007) [20] reference pattern, indicating that the protein isolated from chayote seed by alkaline or UAE possesses a good protein quality to be used as a nutraceutical.

#### 3.4.2. Protein Quality 

The protein quality of CSp and CSPIs was measured in terms of EAA scores and digestibility. The EAA score is based on the proportion of EAA within the daily requirements provided by FAO/WHO/UNU (2007) [20]; values < 1 correspond to limiting EAA. According to Table 2, Trp and Leu were the limited amino acid found in CSp, while CSPIs were only limited in Leu. Vinayashree and Vasu (2021) [2] indicated Lys, Thr, and His as the limiting EAAs in proteins isolated from pumpkin seeds. Also, Wani et al. (2011) [34] reported limiting levels of Lys in protein isolated from watermelon seeds. Regarding EAAI, which is considered an adequate balance of the whole integrity of EAAs, CSp (103.03%), and CSPIs (153.58–424.25%) reported values > 100, showing an increased trend within the protein content (5.5% dw to 8.2% dw, as depicted in Table 1). In this context, the CSPIs obtained by alkaline and ultrasound treatments may perfectly meet the quality expectations of AAS from FAO/WHO/UNU (2007) [20] and EAAI, allowing nutritional interest as a new plant-based protein. The highest EAAI observed for CSPI.UAE-20 kHz (424.25%) indicates that UAE by a probe device (20 kHz, 500 W, 25% amplitude) is greatly efficient in providing chayote seed protein isolates with good quality. Concerning protein digestibility, as shown in Table 3, CSp presented a significantly lower (*p* < 0.05) protein digestibility (47.2%) compared to CSPIs, either obtained by alkaline (76.8%) or ultrasound treatments (80.3% and 86.1% for UAE assisted by a prove or a water bath, respectively). To evaluate the protein quality after in vitro digestion, FAO/WHO/UNU (2007) [20] recommends the use of protein digestibility corrected amino acids score (PDCAAS). It was shown that the PDCAAS from CSPI.UAE-20 kHz (73.6 ± 3.3) and CSPI.UAE-40 kHz (86.1 ± 3.6) was significantly higher (*p* < 0.05) than that of CSp (24.1 ± 2.8). 

### 3.5. Biological Properties of CSPIs

#### 3.5.1. Total Phenolic Content and Antioxidant Activity

As shown in Figure 6A, the mean total phenolic content (TPC) of CSp was 2.65 ± 0.28 mg GAE/g dw, which was half lower than the values found by Ordoñez et al. (2003) [35] for chayote seed extracts prepared by maceration with ethanol for seven days at room temperature (5 mg quercetin equivalents/g dw). When submitted to alkaline and ultrasounds treatments, the TPC increased by two-fold and four-fold, respectively. No differences in TPC values were observed when UAE was performed with a probe (CSPI.UAE-20 kHz, 7.22 ± 0.47 mg GAE/g dw extract) or a water bath apparatus (CSPI.UAE-20 kHz, 7.45 ± 0.85 mg GAE/g dw extract). These findings agree with other works reporting that UAE co-extracted polyphenols with proteins from other seed materials [36]. Although the UAE showed a high effect on the phenolic extraction from chayote seeds, levels present in CSPIs did have a negative impact on their functional properties, as already discussed (Figure 3, Figure 4 and Figure 5).

The in vitro antioxidant properties of CSp and CSPIs were evaluated by reducing power and ABTS radical scavenging activity assays. As shown in Figure 6B,C, both activities increased in a concentration-dependent manner, being higher for the concentration tested at 100 μg/mL. At this concentration, the CSp exhibited an ABTS radical scavenging activity reduction of 28 ± 4.9%, which was two-fold lower than values observed for protein isolates obtained by alkaline and ultrasound treatments. The same pattern was observed for reducing power; with the highest value found for CSPI.UAE-20 kHz and CSPI.UAE-40 kHz. These results indicate that the cavitation effect produced by ultrasounds could significantly reduce the size of protein, change the structure of protein, expose more hydrophobic groups, and provide more free radical reaction sites [8], thus improving the antioxidant activities of chayote seed proteins. It is well documented in the literature that aromatic amino acids (Tyr, His, Trp, and Phe) could enhance the scavenging of free radicals by providing protons [37]. In this context, and as previously discussed (Table 2), CSPI.UAE-20 kHz and CSPI.UAE-40 kHz are rich in Tyr and Phe, as well as the hydrophobic amino acids Asp and Leu, which could contribute to the better free radical scavenging ability. These results were consistent with observations for other proteins isolated from Cucurbitaceae seeds [2,3,37]. Moreover, the presence of phenolics compounds (Figure 6A) could explain the antioxidant activities observed for CSp and CSPIs. It was observed a positive correlation between TPC and ABTS radical scavenging activity (*r*^2^ = 0.7925), as well as between TPC and reducing power (*r*^2^ = 0.7059). Overall, these results suggest that protein isolates from chayote seeds can be considered promising natural antioxidant sources that are useful in the prevention of free radical-mediated diseases.

#### 3.5.2. Inhibition of α-Amylase

The enzyme α-amylase hydrolyzes the complex starch to oligosaccharides, whereas α-glucosidase hydrolyzes hydrolysis oligosaccharides, trisaccharides, and disaccharides into glucose and other monosaccharides [38]. Therefore, inhibition of these enzymes’ activity is a novel approach to regulating blood glucose levels and treating diabetes. Several α-amylase and α-glucosidase inhibitors have been isolated from the plant material and reported as useful in regulating blood glucose levels. Polyphenols and bioactive peptides have been suggested as responsible for such activity [39]. For instance, identified phenolic compounds in apple seed are well-known for α-glucosidase inhibition and explains their antidiabetic properties [40]. In this study, proteins isolated from chayote seeds demonstrated in vitro α-amylase inhibitory potential, as observed in Figure 6D. The α-amylase inhibition increased in a concentration-dependent manner, being higher for the concentration tested, 100 μg/mL. The range of α-amylase inhibition was 10–22%, 25–55%, 35–74%, and 48–80% for CSp, CSPI.AE, CSPI.UAE-20 kHz, and CSPI.UAE-40 kHz, respectively. At 100 μg/mL, protein isolates obtained by alkaline treatment (CSPI.AE) exhibited lower α-amylase inhibition (*p* < 0.05) than protein isolates obtained by UAE. The highest α-amylase inhibition (*p* < 0.05) displayed by ultrasound extracts (CSPI.UAE-20 kHz and CSPI.UAE-40 kHz) could be related to the highest TPC (Figure 6A) and MW fraction lower than 6.5 kDa (Figure 2B). Thus, it would be interesting to isolate and identify the phenolic and bioactive compounds responsible for the observed activities and conduct sufficient in vivo investigation to extrapolate the use of chayote-seed protein isolates in humans.

## 4. Conclusions

This study provided the first insight into the structural, functional, antioxidant, and antidiabetic potential of unexplored chayote seeds. Overall, the results suggest that ultrasounds are an effective technology to be used in the food industry to greatly contribute to the bio-functional and nutritional properties of chayote seed proteins. UAE significantly improved the protein extraction yield and functional properties (protein solubility, turbidity, and emulsifying and foaming properties) of chayote seeds. This effect was more pronounced in probe sonication (20 kHz) rather than bath sonication (40 kHz) for the same period of treatment, 20 min. The CSPI obtained by UAE-20 kHz contained 8.2 ± 0.9% dw of proteins with a balanced amino acid profile, higher content of essential amino acids (315.63 mg/g of protein), and higher protein digestibility (80.3 ± 4.5%). Furthermore, CSPI.UAE-20 kHz exhibited the highest phenolic content (7.22 mg GAE/g dw), higher antioxidant capacity, and higher anti-diabetic activity (α-amylase inhibition of 74%, at 100 μg/mL concentration), suggesting its potential as a nutraceutical. 

## Figures and Tables

**Figure 1 foods-12-02949-f001:**
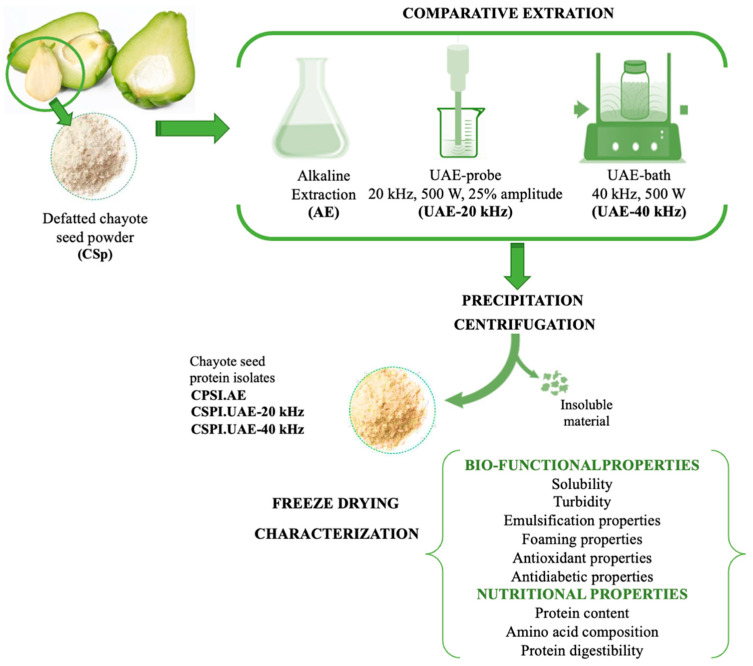
Schematic procedure for preparation of chayote seed protein isolates (CSPIs).

**Figure 2 foods-12-02949-f002:**
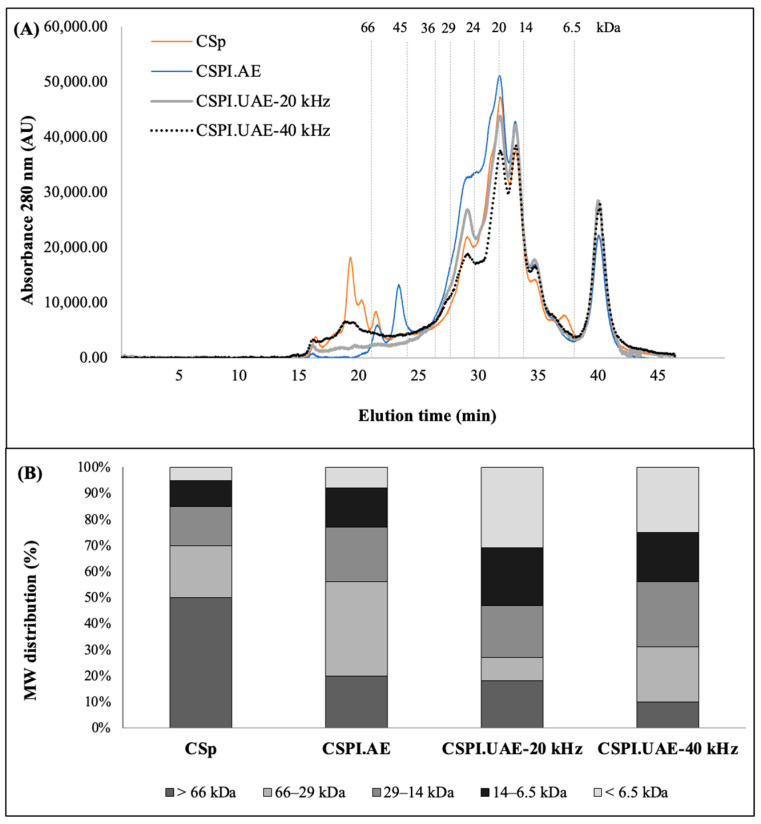
Size exclusion chromatographic profiles (**A**) and molecular weight distribution (**B**) of chayote seed power (CSp) and chayote seed protein isolates produced by alkaline (CSPI.AE) and ultrasound treatments (CSPI.UAE-20 kHz and CSPI.UAE-40 kHz). Samples were analyzed at the same protein concentration (1 mg protein/mL) and injection volume of 20 µL. Absorbance (214 nm) is expressed in arbitrary units (AU). Molecular weight markers: albumin (66 kDa), ovalbumin (45 kDa), glyceraldehyde-3-phosphate dehydrogenase (36 kDa), carbonic anhydrase (29 kDa), trypsinogen (24 kDa), trypsin inhibitor (20 kDa), α-lactalbumin (14.2 kDa), and aprotinin (6.5 kDa).

**Figure 3 foods-12-02949-f003:**
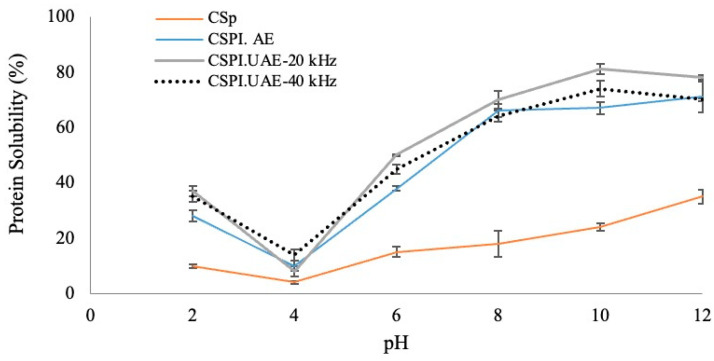
Protein solubility (%) of chayote seed power (CSp) and chayote seed protein isolates produced by alkaline (CSPI.AE) and ultrasound treatments (CSPI.UAE-20 kHz and CSPI.UAE-40 kHz). Values are presented as mean ± standard deviation (*n* = 3).

**Figure 4 foods-12-02949-f004:**
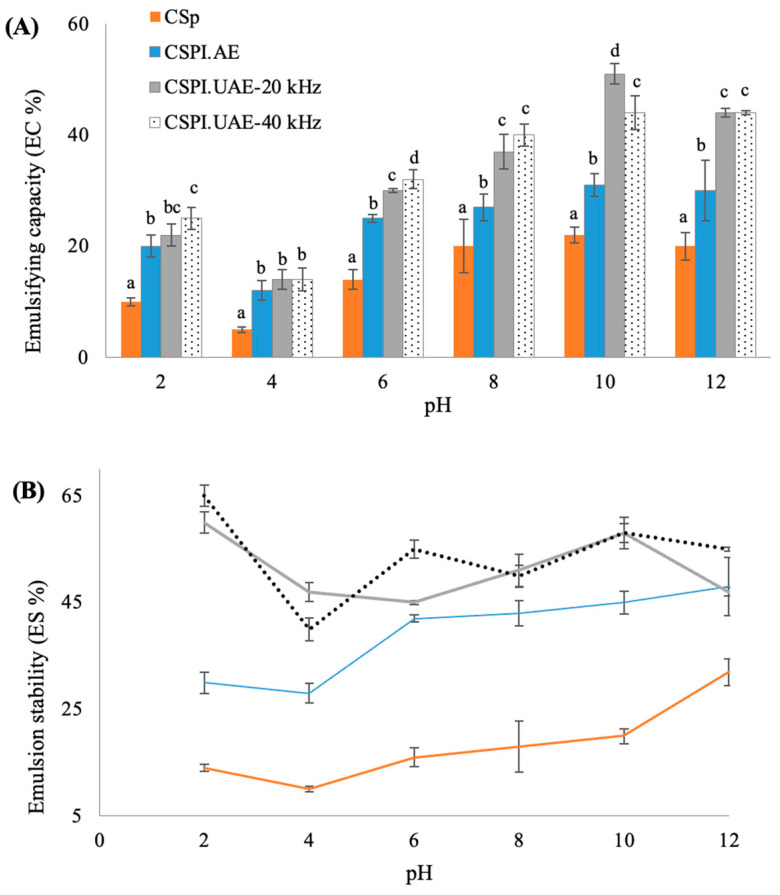
Effect of pH on (**A**) emulsifying capacity (EC %) and (**B**) emulsion stability (ES %) of chayote seed power (CSp) and chayote seed protein isolates produced by alkaline (CSPI.AE) and ultrasound treatments (CSPI.UAE-20 kHz and CSPI.UAE-40 kHz). Values are presented as mean ± standard deviation (*n* = 3). For each pH, different letters show significant differences (*p* < 0.05) between groups (Duncan test).

**Figure 5 foods-12-02949-f005:**
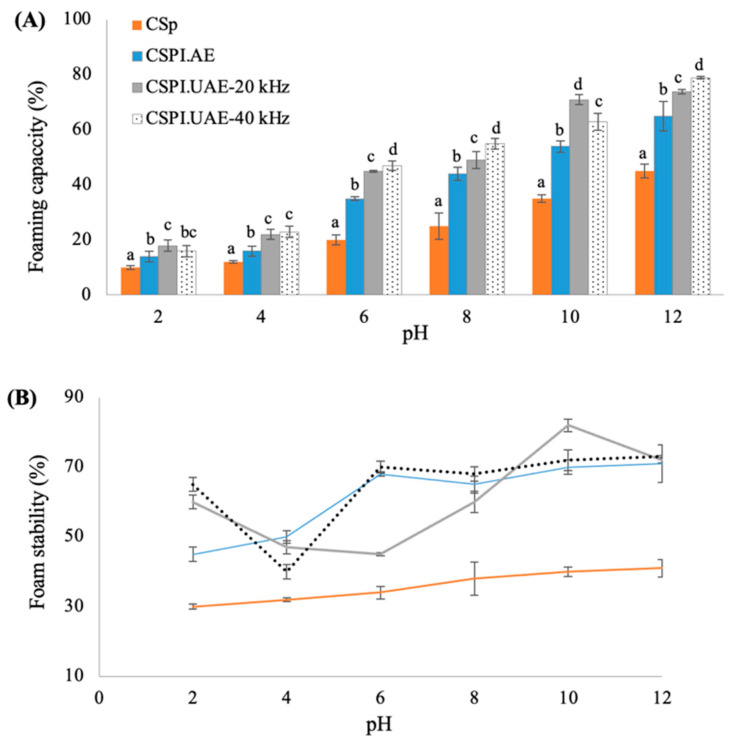
Effect of pH on (**A**) foaming capacity (%) and (**B**) foam stability (%) of chayote seed power (CSp) and chayote seed protein isolates produced by alkaline (CSPI.AE) and ultrasound treatments (CSPI.UAE-20 kHz and CSPI.UAE-40 kHz). Values are presented as mean ± standard deviation (*n* = 3). For each pH, different letters show significant differences (*p* < 0.05) between groups (Duncan test).

**Figure 6 foods-12-02949-f006:**
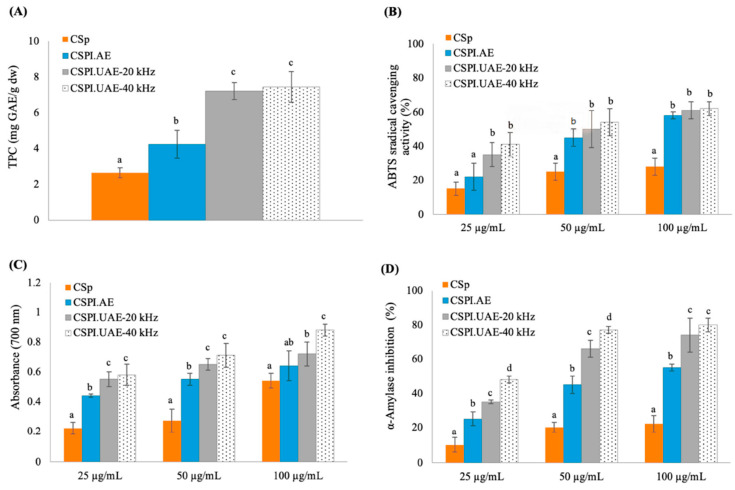
Total phenolic content, TPC (**A**), Reducing power (**B**), ABTS radical scavenging activity (**C**), and α-Amylase inhibition (**D**) of chayote seed power (CSp) and chayote seed protein isolates produced by alkaline (CSPI.AE) and ultrasound treatments (CSPI.UAE-20 kHz and CSPI.UAE-40 kHz). Values are presented as mean ± standard deviation (*n* = 3). For each extract concentration (25, 50 and 100 µg/mL) different letters show significant differences (*p* < 0.05) between groups (Duncan test).

**Table 1 foods-12-02949-t001:** Percentage of protein content (% dw), protein yield recovery (%), and protein purity (%) obtained for different chayote seed protein isolates (CSPIs).

	Protein (% dw)	Protein Yield Recovery (%)	Protein Purity (%)
CSp ^1^	5.5 ± 0.4 ^a^	-	-
CSPI.AE ^2^	6.6 ± 0.6 ^b^	58.0 ± 2.2 ^a^	47.9 ± 1.8 ^a^
CSPI.UAE-20 kHz ^3^	8.2 ± 0.9 ^d^	88.0 ± 3.2 ^c^	66.7 ± 2.5 ^b^
CSPI.UAE-40 kHz ^4^	7.7 ± 1.3 ^c^	72.4 ± 5.6 ^b^	64.4 ± 5.1 ^b^

^1^ Chayote seed power; ^2^ Chayote seed protein isolate obtained by alkaline extraction; ^3^ Chayote seed protein isolate obtained by ultrasound probe (20 kHz, 500 W, 25% amplitude); ^4^ Chayote seed protein isolate obtained by ultrasound bath (40 kHz, 500 W); Values are presented as mean ± standard deviation (*n* = 3). Different letters in the same column show significant differences (*p* < 0.05) between groups (Duncan test).

**Table 2 foods-12-02949-t002:** Amino acid composition (mg/g of protein) of chayote seed power (CSp) and chayote seed protein isolates produced by alkaline (CSPI.AE) and ultrasound treatments (CSPI.UAE-20 kHz and CSPI.UAE-40 kHz), and amino acid scores (AAS) with respect to the provisional scoring pattern of FAO/WHO/UNU (2007).

AA	CSp	CSPI.AE	CSPI.UAE-20 kHz	CSPI.UAE-40 kHz	FAO/WHO/UNU (2007) [20]
Asp	34.70	45.34	57.83	54.32	
Glu	65.70	65.50	87.37	83.37	
Ser	31.10	41.88	66.33	66.33	
Thr	23.50 ^(1.0)^	25.39 ^(1.1)^	32.84 ^(1.4)^	31.14 ^(1.4)^	23
His	15.15 ^(1.0)^	15.44 ^(1.0)^	17.90 ^(1.2)^	15.52 ^(1.0)^	15
Gly	12.50	16.80	27.82	28.21	
Gln	17.40	21.93	37.43	33.08	
Asn	8.70	19.50	21.47	21.46	
Arg	135.40	147.27	167.46	166.95	
Ala	25.40	29.73	38.79	30.91	
Tyr	18.10	20.30	32.55	31.85	
Lys	46.80 ^(1.0)^	47.62 ^(1.1)^	48.97 ^(1.1)^	48.70 ^(1.1)^	45
Val	40.80 ^(1.0)^	43.70 ^(1.1)^	47.85 ^(1.2)^	47.42 ^(1.2)^	39
Met	7.57	8.00	8.50	8.70	
Trp	5.60 ^(0.9)^	6.01 ^(1.0)^	6.15 ^(1.0)^	6.05 ^(1.0)^	6
Cys	25.02	28.02	31.25	30.25	
Phe	20.60	31.35	44.71	40.13	
Ile	31.20 ^(1.0)^	33.70 ^(1.1)^	49.80 ^(1.7)^	47.76 ^(1.6)^	30
Leu	30.10 ^(0.5)^	32.56 ^(0.6)^	58.91 ^(0.9)^	54.06 ^(0.9)^	59
Pro	20.65	28.86	21.20	20.90	
∑ AA ^1^	615.97	708.91	905.14	867.09	
∑ EAA ^2^	221.31	243.77	315.63	299.46	
∑ AAA ^3^	38.70	51.65	77.26	71.98	
∑ SAA ^4^	32.57	36.02	39.75	38.95	
∑ FAA ^5^	138.30	157.37	211.82	196.81	
% EAA	35.93	34.39	34.87	34.54	
% FAA	22.45	22.20	23.40	22.70	
EAAI (%) ^6^	103.03	153.58	424.25	339.29	

^1^ AA, total amino acids; ^2^ EAA, essential amino acids (Thr, Val, Met, Ile, Leu, Phe, His, Lys, and Trp); ^3^ AAA, aromatic amino acids (Phe, Tyr); ^4^ SAA, sulfur amino acids (Met and Cys); ^5^ FAA, flavour amino acids (Asp, Glu, Gly, and Ala); ^6^ EAAI, essential amino acid index.

**Table 3 foods-12-02949-t003:** In vitro protein digestibility and protein digestibility corrected amino acid score (PDCAAS) of chayote seed power (CSp) and chayote seed protein isolates produced by alkaline (CSPI.AE) and ultrasound treatments (CSPI.UAE-20 kHz and CSPI.UAE-40 kHz).

	Protein Digestibility (%)	PDCAAS ^5^
CSp ^1^	47.2 ± 3.1 ^a^	24.1 ± 2.8 ^a^
CSPI.AE ^2^	76.8 ± 3.1 ^b^	42.4 ± 1.7 ^b^
CSPI.UAE-20 kHz ^3^	80.3 ± 4.5 ^bc^	73.6 ± 3.3 ^c^
CSPI.UAE-40 kHz ^4^	86.1 ± 1.4 ^c^	86.1 ± 3.6 ^d^

^1^ Chayote seed power; ^2^ Chayote seed protein isolate obtained by alkaline extraction; ^3^ Chayote seed protein isolate obtained by ultrasound probe (20 kHz, 500 W, 25% amplitude); ^4^ Chayote seed protein isolate obtained by ultrasound bath (40 kHz, 500 W); ^5^ Protein digestibility corrected amino acid score (PDCAAS). Values are presented as mean ± standard deviation (*n* = 3). Different letters in the same column show significant differences (*p* < 0.05) between groups (Duncan test).

## Data Availability

The data used to support the findings of this study can be made available by the corresponding author upon request.

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
