# Peer review of "Chayote (Sechium edule (Jacq.) Swartz) Seed as an Unexploited Protein Source: Bio-Functional and Nutritional Quality of Protein Isolates"

_foods, 2023, doi:10.3390/foods12152949_

Round 1
Reviewer 1 Report
Comments and Suggestions for Authors
The manuscript deals about the use of different methods to evaluate and improve the extraction of proteins from chayote seeds, a source of vegetable proteins not explored as an alternative source of animal origin. Thus, the aim of the work is to evaluate the effect of alkaline extraction and the use of ultrasound (comparing water bath and probe devices) treatments on the recovery of protein from chayote seeds. The protein isolates were then characterized in relation to their bioactivities and techno-functional properties as well as their bioavailability.
The manuscript is complete and is very well written. The methodology is adequate and the results are consistent and are solidly discussed and justified. The conclusions obtained are correlated with the proposed objectives.
Just a few small considerations to improve the work:
1- In section 2.1 (reagents) from the materials and methods, the authors must identify the source of the α-amylase used both in the digestibility tests and in the enzymatic inhibition tests (inhibition of α-amylase).
2- The authors must indicate the E/S relationship used in the digestibility assays (for the oral, gastric and intestinal digestive steps)
3- The α-amylase inhibition assay was carried out according to the iodine-starch method. Why the authors used the iodine methodology to evaluate the inhibition of a-amylase instead of the DNS reagent, commonly used in these determinations? Please indicate how many enzyme units were used in the tests.
3- Why the authors did not also perform the inhibitory activity of α-glucosidase to evaluate the in vitro antidiabetic activity.
These suggestions could improve the manuscript.
Author Response
#1. The manuscript deals about the use of different methods to evaluate and improve the extraction of proteins from chayote seeds, a source of vegetable proteins not explored as an alternative source of animal origin. Thus, the aim of the work is to evaluate the effect of alkaline extraction and the use of ultrasound (comparing water bath and probe devices) treatments on the recovery of protein from chayote seeds. The protein isolates were then characterized in relation to their bioactivities and techno-functional properties as well as their bioavailability. The manuscript is complete and is very well written. The methodology is adequate and the results are consistent and are solidly discussed and justified. The conclusions obtained are correlated with the proposed objectives. Just a few small considerations to improve the work. These suggestions could improve the manuscript.
Answer: Authors thank the appreciation comments of the reviewer; efforts were made to address all the valuable comments to improve the quality of manuscript.
#2. In section 2.1 (reagents) from the materials and methods, the authors must identify the source of the α-amylase used both in the digestibility tests and in the enzymatic inhibition tests (inhibition of α-amylase).
Answer: Pancreatic α-amylase (Sigma-A3176) was used in both assays. This information was added in Line 92.
#3. The authors must indicate the E/S relationship used in the digestibility assays (for the oral, gastric and intestinal digestive steps)
Answer: Authors thank this valuable suggestion; the information was included in Lines 232-239, as follows: “Oral (1 g of each sample was mixed with 4 mL of simulated salivary fluid (SSF) containing 0.5 mL of α-amylase (75 U/mL) at pH 7 for 2 min of incubation), gastric (each oral digest was mixed with 8 mL of simulated gastric fluid (SGF) containing pepsin (2,000 U/mL) at pH 3 for 2 h of incubation), and intestinal (each gastric digest was mixed with 8 mL of simulated intestinal fluid (SIF) containing pancreatin (100 U/mL) and bile (10 mM) at pH 7 for 2 h of incubation) digestive mixtures were incubated in a water bath at 37 ºC and constant shaking (100 rpm). SSF, SGF and SIF solutions were prepared according to Minekus et al. (2014) [18].
#4. The α-amylase inhibition assay was carried out according to the iodine-starch method. Why the authors used the iodine methodology to evaluate the inhibition of a-amylase instead of the DNS reagent, commonly used in these determinations? Please indicate how many enzyme units were used in the tests.
Answer: The α-amylase inhibition assay was carried out according to the well-stablished method of Mao and Kinsella (1981), which is based on starch-iodine colour changes and uses hydrochloric acid (https://doi.org/10.3390/molecules27072307,https://doi.org/10.3390/molecules24244442) or DNS reagent (https://doi.org/10.3390/separations9080190;https://doi.org/10.1016/j.foodchem.2022.134434; https://doi.org/10.1016/j.foodchem.2015.05.120) to stop the reaction of iodine solution with a-amylase. We decided to follow the Esfandi at al. (2022) protocol (https://doi.org/10.1111/jfbc.13762), that uses 1 M hydrochloric acid to stop the reaction. The information about the enzymes unit was included in Line 310, as follows: “… 10 μL of α-amylase (15 U/mL in PBS solution).”.
#5. Why the authors did not also perform the inhibitory activity of α-glucosidase to evaluate the in vitro antidiabetic activity.
Answer: authors agree with reviewer that performing the in vitro inhibitory of other anti-diabetic enzymatic markers (α-glucosidase, and dipeptidyl peptidase IV) should better elucidate the potential anti-diabetic properties of the formulated chayote seed protein isolates. However, authors were not able to conduct these assays due some laboratory logistic management (unavailable α-glucosidase and DPP-IV standards). Still, this was the first screening of the anti-diabetic property in this kind of vegetable protein isolates and like other research works that only assessed the in vitro inhibitory of one anti-diabetic enzymatic marker (α-amylase)
(https://doi.org/10.3390/molecules27072307;https://doi.org/10.3390/molecules24244442;https://doi.org/10.3390/separations9080190;https://doi.org/10.1016/j.foodchem.2022.134434;https://doi.org/10.1016/j.foodchem.2015.05.120),
authors are aware of the limitations of only performing the in vitro assays to screen this biological activity. For this reason, authors had already stated in Lines 623-625 that “… it would be interesting isolate and identify the phenolic and bioactive compounds responsible for the observed activities and conduct sufficient in vivo investigation to extrapolate the use of chayote-seed protein isolates in humans.”

Reviewer 2 Report
Comments and Suggestions for Authors
MS # foods-2521448
Chayote (Sechium edule (Jacq.) Swartz) seed as an unexploited protein source: bio-functional and nutritional quality of protein isolates
This is an interesting study on evaluating the protein quality, nutritional, biological, and functional properties of chayote seed protein extracted using alkaline extraction assisted by ultrasound treatment. Overall, the manuscript was well written. The reviewer would suggest minor revisions.
Specific comments:
The production yield of chayote is suggested to be added in the introduction section.
Section 2.4: what is protein purity?
Table 1: why does further increasing the strength of ultrasound treatment from 20 kHz to 40 kHz reduced protein extraction?
Figure 2, it was expected that proteins would be of small molecular weight in UAE-40kHz > UAE-20kHz, just like the proportion of > 66kD became smaller after UAE-40kHz treatment than after 20kHz; however, UAE-40kHz treatment did not further increase the proportion of < 6.5 kD.
Please include responses to these questions in the discussion in more detail.
Comments on the Quality of English Language
Minor check is needed
Author Response
#6. This is an interesting study on evaluating the protein quality, nutritional, biological, and functional properties of chayote seed protein extracted using alkaline extraction assisted by ultrasound treatment. Overall, the manuscript was well written. The reviewer would suggest minor revisions. Please include responses to these questions in the discussion in more detail.
Answer: Authors thank the appreciation comments of the reviewer. Efforts were made to consider all valuable suggestion by the reviewer.
#7. The production yield of chayote is suggested to be added in the introduction section.
Answer: Authors thank this valuable suggestion; this information was included in Lines 67-69, as follows: “This Cucurbitacea specie grows in tropical, subtropical, and warm regions around the world and each plant can produce 80-100 fruits, reaching production yields of 54,000 lb/ac [10].”. A new reference was included in the reference list.
#8.Section 2.4: what is protein purity?
Answer: The protein purity of CSPIs were, respectively, 47.9 ± 1.8, 66.7 ± 2.5 and 64.4 ± 5.1% for chayote seed protein isolates obtained by alkaline and ultrasonic treatments. These results are now presented in Table 1 and related information was included in Lines 151-161 and 328-334.
#9.Table 1: why does further increasing the strength of ultrasound treatment from 20 kHz to 40 kHz reduced protein extraction?
Answer: To better clarify the effect of UAE treatment (probe-20 kHz and water bath- 40 kHz) on protein extraction from CSp, the discussion was improved in Lines 328-334, as follow: “In addition, the ultrasonic probe device (20 kHz) favoured the direct contact between tip and sample, which do not occur in water bath apparatus (40 kHz), where samples do not contact directly with irradiating surface. Thus, CSp treated with 20 kHz probe is more exposed to high power compared to 40 kHz water bath, leading to greater extraction efficiency. Moreover, the ultrasound treatment produced CSPIs with higher (p < 0.05) protein purity, the values found were 66.7 ± 2.5% and 64.4 ± 5.1 for CSPI.UAE-20 kHz and CSPI.UAE-40 kHz, respectively.”.
#10. Figure 2, it was expected that proteins would be of small molecular weight in UAE-40kHz > UAE-20kHz, just like the proportion of > 66kD became smaller after UAE-40kHz treatment than after 20kHz; however, UAE-40kHz treatment did not further increase the proportion of < 6.5 kD.
Answer: Please see the previous comment. New information was added to Lines 363-365, as follow: “CSp treated with 20 kHz probe is more exposed to high power compared to 40 kHz water bath, leading to greater hydrolysis of proteins with formation of smaller peptides.”.
